# Cancer-Associated Fibroblasts Genes and Transforming Growth Factor Beta Pathway in Gastric Cancer for Novel Therapeutic Strategy

**DOI:** 10.3390/cancers17050795

**Published:** 2025-02-26

**Authors:** Hiroyuki Minoura, Riku Okamoto, Naoki Hiki, Keishi Yamashita

**Affiliations:** 1Division of Advanced Surgical Oncology, Research and Development Center for New Medical Frontiers, Kitasato University School of Medicine, Kitasato 1-15-1, Minami-ku, Sagamihara, Kanagawa 252-0374, Japan; minoura.hiroyuki@kitasato-u.ac.jp (H.M.); okamoto.riku@kitasato-u.ac.jp (R.O.); 2Department of Upper Gastrointestinal Surgery, Kitasato University School of Medicine, Kitasato 1-15-1, Minami-ku, Sagamihara, Kanagawa 252-0374, Japan; nhiki@med.kitasato-u.ac.jp

**Keywords:** CAFs, gastric cancer, prognosis, therapeutic strategy

## Abstract

Gastric cancer remains one of the leading causes of cancer-related deaths worldwide, and its progression is strongly influenced by the tumor microenvironment, particularly cancer-associated fibroblasts (CAFs). These fibroblasts contribute to tumor growth, metastasis, and resistance to treatment. This study focuses on identifying specific genes associated with CAFs and their role in gastric cancer. By analyzing gene expression patterns through single-cell RNA sequencing, we found that several key genes, including *SPARC*, *THBS2*, *COL1A1*, and *INHBA*, are linked to poor prognosis in gastric cancer. Interestingly, not all these genes are restricted to CAFs alone, suggesting they may also reflect broader tumor-stroma interactions. Moreover, the interaction between CAFs-associated genes and the transforming growth factor beta (TGF-β) pathway appears to play a critical role in cancer progression. Understanding these molecular mechanisms could lead to new treatment strategies that target the tumor microenvironment, improving outcomes for patients with gastric cancer.

## 1. Introduction

Gastric cancer (GC) is the fifth most common cancer [1]. Its molecular subtypes, as classified by The Cancer Genome Atlas (TCGA), include microsatellite-instable GC (MSI GC), representing DNA hypermutation; Epstein–Barr virus GC (EBV GC); genomic-stable GC (GS GC); and chromosomal-instable GC (CIN GC). These subtypes can potentially affect therapeutic strategies [2]. In GC, immune checkpoint inhibitors (ICIs) are considered effective for MSI GC subtypes (KEYNOTE-158 trial) [3], like their efficacy in colorectal cancer (CRC) (KEYNOTE-164 trial) [4].

GC is characterized by genetic abnormalities, with significant recurrent mutations identified in *TP53* (50%), *ARID1A* (14%), *PIK3CA* (12%), and *CDH1* (11%), while other genes show mutation rates below 10%. Among the 215 non-hypermutated tumors [2], *TP53* mutations were enriched in CIN GC (71% mutated), while *PIK3CA* mutations were uniquely observed in EBV GC (80% mutated), and *CDH1* mutations were predominantly found in GS GC, particularly in diffuse-type GC (37% mutated). This finding suggests that there are specific characteristics unique to the individual GC molecular subtypes.

Germline mutations in *CDH1* are linked to hereditary diffuse-type GC [5,6]. However, germline analysis revealed only two *CDH1* polymorphisms, neither of which is known to be pathogenic [2]. Hot-Net analysis of MSI GC tumors revealed common alterations in class I HLA genes, including *B2M* and *HLA-B*. *B2M* mutations in other cancers result in the loss of expression of class I HLA genes [7,8], suggesting that these alterations may benefit hypermutated cases by reducing antigen presentation to the immune system.

Secretome factors from cancer-associated fibroblasts (CAFs) have recently garnered attention for their role in cancer initiation and/or progression in CRC, pancreatic adenocarcinoma (PDAC), and breast cancer (BC) [9,10,11]. In GC, *keratinocyte growth factor* (*KGF*/*FGF7*) was the first to be shown to be secreted by gastric CAFs, although it was not expressed in GC tumor cells [12]. Interestingly, scirrhous GC, the most malignant subtype of GC, expressed the *KGF receptor* (*FGFR2*) at higher levels than other GC subtypes, indicating that the FGF7-FGFR2 axis might be critical for scirrhous GC progression.

The functional relevance of *TGF-β* signaling in CAFs was examined in GC through the conditional inactivation of the *TGF-β type II receptor* (*TGFBR2*) gene in mouse fibroblasts [13]. The loss of TGF-β responsiveness in fibroblasts led to intraepithelial neoplasia in the forestomach, associated with an increased abundance of stromal cells, wherein the activation of paracrine *hepatocyte growth factor* (*HGF*) signaling was identified as one potential mechanism for stimulating epithelial proliferation (Figure 1).

This figure illustrates the representative interactions among gastric cancer cells (red), cancer-associated fibroblasts (CAFs; blue), pericytes (green), and immune cells, including myeloid-derived suppressor cells (MDSCs; yellow) and other myeloid cells (purple), within the tumor microenvironment (TME). It highlights potential therapeutic targets and key signaling pathways. Pericytes were more obvious than CAFs in GC than in CRC. In our scRNA-seq, they strongly expressed *ASPN*, *ACTA2*, and *PDGFRB*. *ASPN*, secreted by pericytes, promotes GC cell invasion via paracrine effects mediated by the CD44-Rac1 axis. This suggests that *ASPN* could be a potential therapeutic target for TME modulation, in addition to *ACTA2*. *INHBA*, secreted by myeloid cells and CAFs, induced *FAP* on CAFs, which may affect GC cells (INHBA-FAP axis). CAFs secrete *WNT5A* and *POSTN*, which activate the *FZD* receptor on GC cells. This activation induces the *ERK* pathway, enhancing cancer cell proliferation and survival (WNT4A-FZD-ERK axis). Impaired TGFβ signaling, caused by reduced *TGFBR2* expression in CAFs, leads to increased secretion of *HGF*, further contributing to cancer progression (TGFB-HGF axis). Pericytes and CAFs secrete PDGF family ligands (*PDGFA*, *PDGFB*, *PDGFC*), which interact with PDGFR on stromal cells. Additionally, CAFs-derived CXCL chemokines (*CXCL1*, *CXCL3*, *CXCL5*, *CXCL8*) recruit MDSCs via *CXCR1*/*CXCR2*, promoting tumor immune evasion. Blocking *PDGFR* is proposed as a potential therapeutic strategy (PDGFs-CXCLs axis).

These pathways underscore the dynamic and reciprocal interactions between stromal and cancer cells within the TME, identifying novel therapeutic approaches for gastric cancer treatment. Black arrows with green secretion factors indicate the activation or stimulation of signaling pathways. Increased gene or protein expression is indicated by red arrows, while decreased expression is indicated by blue arrows. Blocked arrows (⊥) represent pathway disruptions.

***ACTA2****: Alpha Smooth Muscle Actin 2—A marker of activated fibroblasts. **INHBA**: Inhibin Subunit Beta A—A growth factor involved in tumor progression. **PDGFRB**: Platelet-Derived Growth Factor Receptor Beta—A receptor involved in stromal activation and angiogenesis. **FAP**: Fibroblast Activation Protein—A protease expressed in activated fibroblasts. **ASPN**: Asporin—A protein that regulates collagen and tumor invasion. **PDGFC**: Platelet-Derived Growth Factor C—A growth factor involved in cell proliferation. **POSTN**: Periostin—A protein that supports cancer metastasis. **Rac1**: A small GTPase involved in cytoskeletal organization and cell migration. **CD44**: A cell surface receptor involved in cell adhesion and migration. **FZDs**: Frizzled Family Receptors–A family of receptors mediating Wnt signaling. **ERKs**: Extracellular Signal-Regulated Kinases—Proteins involved in cell proliferation and survival. **TGF-β**: Transforming Growth Factor Beta—A cytokine involved in cell differentiation and immune regulation. **CXCL**: C-X-C Motif Chemokine Ligand—Chemokines involved in immune cell recruitment. Examples: CXCL1, CXCL3, CXCL5, CXCL8*.

We recently defined CAFs-associated genes (CAFGs) in CRC based on (1) a close association (R = 0.9 or greater, underlined genes throughout this paper, such as *FGF7* and *TGFBR2*) with the expression of *SPARC*, a well-known stromal marker, in the cancer stroma of CRC tumors (GSE35607), and (2) stromal specificity (stromal/epithelial expression ratio, SE ratio = 10 or greater), similar to *SPARC* (SE = 17.2) [9]. In CRC, the 115 CAFGs with the highest expression levels (n = 115) included *FAP* (SE = 20.2), *ACTA2* (SE = 20.2), and *VIM* (SE = 17.8), which are well-known classical CAFs markers [14]. This suggests that the stromal expression of CAFGs may be well synchronized with classical CAFs markers in CRC.

CAFGs were further classified into three categories based on stroma specificity representing SE ratio: CAFGs with a high SE ratio (10 or greater), semi-CAFGs with a middle SE ratio (5 or greater but below 10), and L-CAFGs with a low SE ratio (below 5). For example, *TGFB1* is categorized as a semi-CAFG, distinct from conventional CAFGs (SE ratio = 10 or beyond) [9]. The SE ratio demonstrates stromal specificity, as high expression of individual genes in tumor cells theoretically reduces the SE ratio; a high SE ratio may reflect stroma-specific molecular status, while a low SE ratio indicates their expression status in tumor cells. SE ratio may represent unique cell populations.

In this review article, CAFGs identified in CRC as representative of human cancer were initially explored as candidate CAFGs for GC through a literature search (designated as GC CAFGs throughout this paper), based on the hypothesis that stromal components of different cancers are shared across the human body. Furthermore, we confirmed their expression related to CAFs using single-cell RNA sequencing (scRNA-seq) data from the public database of GC (GSE183904) [15].

## 2. Materials and Methods

### 2.1. Expression Profiles of the Microdissection Tissues of the 13 CRC Tumors

The public databases of stromal (cancer stroma) expression after microdissection of the 13 CRC tumors (GSE35602) were used in the microarray (Human Genome, Whole 4 × 44 K, Agilent Inc., Santa Clara, CA, USA) harboring 45,015 genes [16], where cancer stromal/epithelial expression ratio (SE ratio) was calculated. Signal intensities were normalized using a single *GAPDH* probe (A_23_P13899). Correlation coefficients between genes were calculated for the entire gene set using the “CORRELATION” command in Microsoft^®^ Excel^®^ for Microsoft 365 MSO (version 2501, build 16.0.18429.20132, 32-bit, Microsoft Corporation, Redmond, WA, USA).

### 2.2. scRNA-Seq Analysis

The analysis was conducted using the GC public database of primary GC tumors (GSE183904, n = 26) and compared with the CRC public database (GSE200997, n = 16) [17]. The breakdown of GC stages is as follows: 26 primary lesions from GC patients (Stage I/II/III/IV; 3/6/14/3) [15]. The single-cell RNA sequencing (scRNA-seq) analysis was performed using Seurat (version 4.3.0), implemented in R (version 4.4.2, R Foundation for Statistical Computing, Vienna, Austria). RStudio was used as the integrated development environment (IDE) (available at https://posit.co/download/rstudio-desktop/, accessed on 5 February 2025). The number of genes per cell in the single-cell analysis ranged from 200 to 2500, and only cell populations with a mitochondrial gene ratio below 10% were included. Doublets were removed using Doublet Finder (version 2.0.4), and batch effects were corrected using Harmony (version 1.2.3). As a result, 62,519 cells were finally selected to generate the UMAP.

## 3. Current Research on GC CAFGs and Confirmation of Expressed Cells in scRNAseq Analysis

CAFGs were defined as genes closely (R = 0.9 or greater) associated with *SPARC* based on expression levels in GSE35602 (CRC) [9]. Some of these have also been reported as CAFs-related genes in GC (designated as GC CAFGs). The highest expression levels of such GC CAFGs were confirmed for *SPARC* in CRC, followed by *THBS2* (SE = 30.3), *COL3A1* (SE = 18.5), *COL1A1* (SE = 18.9), *ACTA2*, *INHBA* (SE = 20.9), *PDGFRB* (SE = 14.8), *FAP* (SE = 20.3), *ASPN* (SE = 27.3), *KLF2 *(SE = 10.5), *PDGFC* (SE = 14.6), *GREM1* (SE = 34.1), *FBLN2* (SE = 17.7), *SDC2* (SE = 11.7), *POSTN* (SE = 15.4), *ZEB2* (*ZFHX1B*) (SE = 10.6), *LTBP1* (SE = 10.6), and *FGF7* (SE = 13.4).

The expression levels of the GC CAFGs and their SE ratios in CRC are shown (Figure 2a,b), as microdissection data were not available in GC. GC may not be suitable for microdissection analysis due to the histological features associated with poor differentiation. Hence, this review article on GC will summarize the clinical and functional relevance of CRC CAFGs that have been reported in GC. Representative correlations of individual GC CAFGs (*FAP*, *COL3A1*, *INHBA*, *PDGFRB*, and *POSTN*) with stromal *SPARC* (str*SPARC*) in CRC are shown as representatives in Figure 2c.

In GC, *SPARC* was identified as the most highly expressed gene [18], consistent with findings in CRC (Figure 2a, upper panel), suggesting that it is an excellent standard stromal marker, and it has been repeatedly confirmed to be overexpressed in GC tumor tissues [19]. *SPARC* was predominantly expressed in fibroblasts surrounding tumor cells and endothelial cells [20,21]. Consistent with such reports, scRNA-seq data using the public database of CRC (GSE200997, 16 CRC tumors) confirmed that *SPARC* was expressed in both CAFs and endothelial cells (Figure 2c), with pericytes subcomponents marked by *RGS5* categorized as both CAFs and endothelial cells.

In this review article, we initially confirmed *SPARC* expression through scRNA-seq analysis of GC samples (GSE183904), which included 26 GC tumors [15]. A UMAP was newly constructed (Figure 3a), and each cluster was annotated with individual cell components using unique markers (Figure 3b). In GC, pericytes were clustered independently of CAFs differently from CRC. Thus, these unique cell component patterns may influence tumor phenotypes in GC. As expected, *SPARC* expression was confirmed in both “broad-sense” CAFs (CAFs and pericytes) and endothelia in GC, like CRC (Figure 3c).

## 4. Current Understanding of CAFGs in GC

*SPARC* expression was elevated during GC progression, indicating poor prognosis [18,22]. The prognostic relevance of *SPARC* has been repeatedly confirmed in meta-analyses of GC [23,24], and it has proven useful even in specific clinical situations [25,26,27]. Moreover, recent bioinformatics analyses have recapitulated the hub gene status of *SPARC* among differentially expressed genes (DEGs) in GC [28,29,30].

Among the upregulated DEGs in GC, 10 hub genes, including *SPARC* and *THBS2*, were identified as GC CAFGs, and they were significantly associated with poor overall survival (OS) [31]. KEGG (Kyoto Encyclopedia of Genes and Genomes) pathway analysis indicated that the most important pathway was enriched in extracellular matrix (ECM)-receptor interactions in GC, with hub genes *SPARC* and *THBS2* uniquely identified.

In scRNA-seq, *THBS2* was expressed in subpopulation of CAFs (Figure 3d), and its distribution in CAFs alone of GC was same with CRC tumors (see Figure 2a). The survival chart for *THBS2* indicated an increase in mortality in GC [31], underscoring the importance of *THBS2* in the development of GC and suggesting they could be candidate genes for the prevention and early diagnosis of GC. However, their molecular mechanisms remain elusive in GC.

Recently, bioinformatics software has been utilized to predict hub genes in GC, confirming alternate upregulated significant DEGs and identifying common hub genes. Combined with survival analysis, upregulated genes were identified as central and played important prognostic roles in GC, including CRC CAFGs such as *SPARC*, *COL1A1*, and *COL3A1* [30,31,32,33,34].

Interestingly, our microarray exploration for *SPARC*-target genes (DEGs identified by *SPARC* knockdown) revealed *COL1A1* and *COL3A1* in YS1 cells (GC-derived CAF cells), while forced expression of *SPARC* induced their expression (unpublished). In scRNAseq, both *COL1A1* and *COL3A1* were predominantly expressed in CAFs and pericytes, differently from *THBS2* (Figure 3d). Hence, the SPARC-COL1A1 axis and/or SPARC-COL3A1 axis in “broad-sense CAFs” may have significant prognostic relevance in GC.

Among the classical CAF markers [14], *FAP* and *ACTA2* were also examined in our scRNA-seq (Figure 3d). *FAP* was predominantly expressed in CAFs of GC, while *ACTA2* was mainly expressed in pericytes (pericytes > CAFs). In GC, *FAP* was identified for the first time as a stromal signature associated with poor outcomes [35]. *ACTA2* expression was independently associated with worse OS in a discovery cohort of GC, either validated in three independent cohorts [36].

Our present datasets suggest that *ACTA2* may therefore represent the role of pericytes in GC progression. Interestingly, among GC patients treated with immune checkpoint inhibitors (ICIs), those with low *ACTA2* expression responded better to ICIs compared with those with high *ACTA2* expression [36]. In the scRNA-seq analysis of MSI-positive GC patients treated with ICIs, those who responded to treatment exhibited lower tumor stromal *ACTA2* expression. Therefore, pericytes may play an important role in the responsiveness of GC to ICIs (Figure 1).

Kumar V et al. generated a comprehensive single-cell atlas of GC (over 200,000 cells) across clinical stages and histologic subtypes [15]. They observed increased plasma cell populations in diffuse-type GC and a stage-wise accrual of CAF subpopulations marked by *INHBA* and *FAP* co-expression. Our scRNA-seq data utilized their database, confirming *INHBA* expression in CAFs and myeloid cells (CAFs < myeloid cells, Figure 3d). This distribution pattern was different from that in CRC, where *INHBA* was more dominantly expressed in myeloid cells in GC. Interestingly, *INHBA* robustly induced *FAP* expression in CAFs, and high *INHBA* expression was correlated with poor prognosis in GC [15], suggesting that the INHBA-FAP axis in CAFs might play an important role in GC aggressiveness. Nevertheless, *INHBA* expression in myeloid cells was not accompanied by *FAP* expression, indicating that the prognostic relevance of the INHBA-FAP axis in CAFs remains unclear in GC.

In GC, *platelet-derived growth factor C* (*PDGFC*) and D expressions were significantly associated with poor prognosis, while *PDGF receptor β* (*PDGFRB*) was predominantly expressed in diffuse-type GC stroma [37]. In our scRNA-seq, *PDGFRB* was most dominantly expressed in F-pericytes, unlike *PDGFRA*, while *PDGFC* was weakly expressed in CAFs and myeloid cells, but not in cancer cells (Figure 3d). Importantly, PDGFR blockade and anti-PD1 treatment synergistically suppressed the growth of fibrotic tumors (Figure 1).

In GC, CAFs stimulated with PDGFs exhibited markedly increased expression of *CXCL1*, *CXCL3*, *CXCL5*, and *CXCL8* [37], which are involved in the recruitment of polymorphonuclear myeloid-derived suppressor cells (PMN-MDSCs) that expressed CXCL receptors (*CRCR1*/*CXCR2*). Importantly, *PDGFR* blockade reversed the immunosuppressive tumor microenvironment (TME) through stromal modification, suggesting that MDSCs play a critical role in TME suppression through the PDGFs-PDGFRs axis in GC.

CXCL family genes were expressed in CAFs, but their expression was not restricted to CAFs alone in our data (rather dominantly expressed in myeloid cells > CAFs, Figure 4a). This finding suggests that CXCL family gene expressions rather represent stromal reprogramming, not solely by the PDGFs-CXCL axis in CAFs. Most intriguingly, both *CXCR1* and *CXCR2* were not expressed in myeloid cells of GC (Figure 4b), as well as CRC (Figure 4c), suggesting that MDSCs do not play a critical role in GC progression and CRC.

scRNA-seq also identified four subsets of CAFs within stromal cells defined by *PDGFRA*, *FBLN2*, *ACTA2*, or *PDGFRB*, and each subset was distributed distinctively throughout stomach tissues with varying proportions at each pathologic stage of GC [38]. In our scRNA-seq, both *ACTA2* and *PDGFRB* were expressed dominantly in pericytes, while *PDGFRA* was expressed specifically in CAFs, and no expression of *FBLN2* was found in CAFs (Figure 3d). Among the subsets, the *PDGFRA*+ subset expanded in metaplasia and cancer compared with normal tissue [38]. Intriguingly, the culture of metaplastic gastroids with conditioned media from metaplasia- or cancer-derived fibroblasts also promoted dysplastic transition. In CRC, *PDGFRA* was not included as a CAFG because its expression with strSPARC was below 0.9 (R = 0.78) [9], but it was specifically expressed in CAFs in GC, as described earlier. On the other hand, *FBLN2* was modestly detected in CAFs by our current data (Figure 3d).

In GC, *Asporin* (*ASPN*), a small leucine-rich proteoglycan (SLRP), was reported to be predominantly expressed in CAFs and plays essential roles in promoting co-invasion of CAFs with cancer cells [39]. *ASPN* expression in CAFs was induced by exposure to GC cells [39]. However, our scRNA-seq confirmed that accurate *ASPN* expression was observed predominantly in pericytes (Figure 3d), suggesting that interactions of *ASPN* were performed between pericytes and cancer cells in clinical GC tumors. *ASPN* activates Rac1 via an interaction with *CD44* in CAFs, promoting invasion by CAFs themselves (Figure 4d). Moreover, *ASPN* promotes invasion by neighboring tumor cells through paracrine effects mediated by activation of the CD44-Rac1 axis. Nevertheless, *CD44* is rather modestly expressed in CAFs and pericytes and weakly in cancer cells in comparison to other stromal cells of GC (Figure 4d). Anyway, *ASPN* may represent a new therapeutic target for the development of drugs aimed at manipulating the TME (Figure 1).

The single-cell atlas of GC across clinical stages and histological subtypes identified 34 distinct cell-lineage states, with increased plasma cell proportions discovered in diffuse-type tumors [15]. Previous reports have shown that *KLF2* regulates the homing of plasma cells in multiple myeloma [40], and Kumar et al. therefore examined the hypothesis that the increased recruitment of plasma cells in diffuse-type GC might be regulated by *KLF2* activity. *KLF2* was expressed in many stromal as well as cancer cells in scRNA-seq (Figure 3d). Notably, significant correlations were observed only between plasma cells and *KLF2*-expressing tumor cells, but not between plasma cells and *KLF2*-expressing non-tumor cells, suggesting that *KLF2*-expressing tumor cells may be associated with plasma cell recruitment in diffuse-type GC [15].

CAFs exhibited diverse and distinct DNA methylation and H3K27me3 patterns compared with normal fibroblasts. Loss of H3K27me3, but not DNA methylation, in CAFs was enriched for genes involved in stem cell niches, cell growth, tissue development, and stromal-epithelial interactions, such as *WNT5A*, *GREM1*, and *IGF2* [41], among which *WNT5A* was associated with poor prognosis in GC. Moreover, inhibition of secreted *WNT5A* from CAFs suppressed cancer cell growth and migration (Figure 1). In our scRNA-seq, *WNT5A* was modestly expressed in CAFs and pericytes, as well as *GREM1* (Figure 3d).

TME subtypes are associated with histopathological and genomic characteristics and survival outcomes in GC, with extensive stromal remodeling occurring during GC progression. High *SDC2* expression in CAFs is linked to aggressive phenotypes and poor survival, with *SDC2* overexpression in CAFs contributing to tumor growth [42]. In scRNA-seq, *SDC2* was expressed dominantly in CAFs and pericytes (Figure 3d).

Seven major cell types were identified in the TME of GC, with CAFs, endothelial cells, and myeloid cells categorized as being enriched in the deep layers [43]. Cell-type-specific clustering revealed that the transition from superficial to deep layers was associated with the enrichment of inflammatory endothelial cells, along with upregulated *CCL2* transcripts. Consistent with this, *CCL2* expression was confirmed in endothelia in scRNA-seq (Figure 3d). The elevation of *CCL2* levels along the superficial-to-deep layer axis indicated the presence of immunosuppressive immune cell subtypes that may contribute to tumor cell aggressiveness in the deep invasive layer of diffuse-type GC, and *CCL2* expression correlated with poor survival in GC.

Stromal *periostin* (*POSTN*) was detected at the invasive front of GC tissues and enhanced the in vitro growth of diffuse-type GC cell lines, accompanied by activation of ERK (phosphorylated Erk, designated as Erkp) [44] (Figure 1). Furthermore, co-inoculation of GC cell lines with *POSTN*-expressing NIH3T3 mouse fibroblast cells facilitated tumor formation. Tumors of the gastric wall in *POSTN* (−/−) mice were significantly smaller than those in wild-type mice, with lower *Ki-67* and phosphorylated ERK (pERK) positive rates observed in *POSTN* (−/−) mice. These findings suggested that *POSTN* produced by CAFs constitutes a growth-supportive TME in GC (Figure 1). In scRNA-seq, *POSTN* was predominantly expressed in CAFs (Figure 3d).

*IL33* and its receptor *ST2L* (*IL1RL1*) are upregulated in GC and serve as prognostic markers indicative of poor prognosis for GC patients [45]. CAFs-derived *IL33* enhanced the invasion of GC cells by inducing epithelial-mesenchymal transition (EMT) through the ERK1/2-SP1-ZEB2 axis in a *ST2L*-dependent manner. *ZEB2* was also included among the CRC CAFGs, suggesting that *ZEB2* expression is synchronized with other CAFGs. In scRNA-seq, it was expressed weakly in GC CAFs, and their expressions were not unique to CAFs (Figure 3d), suggesting that *ZEB2* may represent stromal reprogramming rather than a signaling axis in CAFs of GC.

The secretion of *IL33* by CAFs can be induced by the proinflammatory cytokine *tumor necrosis factor alpha* (*TNFA*, SE = 3.2), which is released by GC cells via the TNFR2-NFKB-IRF1 axis [45]. However, in our scRNA-seq, *IL33* was rather predominantly expressed in endothelial cells (>CAFs), and *TNFA* was expressed dominantly in myeloid cells, and T cells (>>cancer cells) (Figure 4e). Given that silencing of *IL33* expression in CAFs inhibited the peritoneal dissemination of GC cells in nude mice, IL33 axis in CAFs may be an important therapeutic target in GC.

The expression of the *IL33* receptor *ST2L* (*IL1RL1*) was confirmed dominantly in mast cells alone (Figure 4e), suggesting that the IL33-ST2L axis is dominant between endothelial cells (>CAFs) and mast cells, therapeutic potential of mast cells on this pathway have remained elusive. Recently, mast cell activation was reported to be critical for Treg mobilization to promote GC progression [46], mediated by *IL33*, which is consistent with our scRNA-seq findings. Our scRNA-seq data proposed the TNFA-IL33-IL1RL1 axis from tumor infiltrates to mast cells, mediated by endothelia and CAFs in GC.

Table 1 summarizes the molecular characteristics of negative prognostic CAFGs in GC, including their SE ratios in CRC, prognostic significance, and expression patterns identified through scRNA-seq analysis. In addition to understanding their molecular roles, we explored therapeutic strategies targeting CAFGs, in which we can potential therapeutic strategies targeting CAFGs in both clinical and preclinical settings. These strategies include sophisticated approaches such as blocking CAFG-related signaling pathways, modulating the tumor microenvironment, and targeting specific CAFG markers identified through single-cell RNA sequencing.

## 5. Current Perspectives of Semi-CAFGs in GC

Semi-CAFGs in CRC were defined by an SE ratio between 5 and 10, alongside an R-index with *SPARC* of 0.9 or greater. Such genes have also been reported in GC, including *TGFB1* (SE = 5.0) and *urokinase-type plasminogen activator* (*PLAU*) (SE = 7.7). Our scRNA-seq clarified that *TGFB1* was ubiquitously expressed in non-specific stromal cells including CAFs but weakly in cancer cells of both CRC and GC (Figure 4f). Therefore, semi-CAFGs defined by SE ratios below 10 may not be guaranteed for CAFs-associated expression of the individual genes like *TGFB1*.

*TGFB1* is essential for invasive growth and proliferation in GC tumor cells, such as *GCTM-1* GC cell line [47], in which *TGFB1* induces enhanced expression of *matrix metalloproteinase 9* (*MMP9*) and *PLAU* in tumor cells. In scRNA-seq, however, *PLAU* was predominantly expressed in CAFs and pericytes, but not in cancer cells (Figure 4g). Moreover, *MMP-9* was not expressed in cancer cells, either, but in myeloid cells. These findings suggested that the TGFB1-MMP9/PLAU axis may not be outstanding in GC tumor cells.

*TGFB1* accelerated the phosphorylation of *SMAD2* (SE = 2.0) and *SMAD3* (SE = 2.1) and the nuclear translocation of the SMAD2/SMAD3-SMAD4 (SE = 2.3) complex in GCTM-1 cells. However, these *TGFB1*-induced effects were significantly inhibited by *IFNG*-induced *SMAD7* (SE = 3.1) expression [47]. When GCTM-1 cells were cotransfected with AS oligonucleotides targeting *SMAD2* and *SMAD3*, the *TGFB1*-induced invasion disappeared. Moreover, the inhibitory effect of *IFNG* on *TGFB1*-dependent GCTM-1 invasion was abolished by transfection with AS oligonucleotides targeting *SMAD7*. Thus, *IFNG* may serve as a therapeutic tool for *TGFB1*-expressing invasive GC (Figure 4h). Nevertheless, the expressions of *SMAD*s were not specifically confirmed in cancer cells by scRNA-seq (non-specific expression). Moreover, *TGFBR2* (SE = 1.6) was expressed dominantly in endothelia, and other stromal cells, but weakly in cancer cells (Figure 4g).

In scirrhous GC cells, administration of the TGFBR inhibitor *A-77* resulted in significantly better prognosis in mice with peritoneal dissemination, as well as a significant decrease in the weight and number of metastatic nodes [48]. *A-77* decreased the expression of integrins *(ITGA2*, SE = 1.3; *ITGA3*, SE = 1.2; and *ITGA5*, SE = 6.4) in GC cells, and histological findings showed a reduced degree of fibrosis in tumors treated with A-77. Although *ITGA5* was classified as a semi-CAFG in GC, it was not expressed in CAFs but rather in endothelial cells and myeloid cells. Its expression is not related to CAFs.

TGFB is released as part of an inactive tripartite complex consisting of TGFB, its propeptide, and a molecule of latent TGFB binding protein (LTBP) (*LTBP1*, SE = 10.6; *LTBP2*, SE = 6.1; *LTBP3*, SE = 2.2; and *LTBP4*, SE = 5.3). Namely, *LTBP1* can be defined as a CAFG (Figure 2a) while *LTBP2* and *LTBP4*as semi-CAFGs. In scRNAseq analysis, *LTBP1* and *LTBP2* were expressed dominantly in CAFs alone (Figure 4g), while *LTBP4* was strongly expressed in CAFs, in addition to pericytes and cancer cells. Thus, *TGFB1* expressed in CAFs is considered initially inactive.

The interaction of TGFB and its cleaved propeptide renders the growth factor latent, and liberation of TGFB from this state is crucial for signaling. Mice with mutations in cysteines that link the propeptide to LTBP, blocking covalent association, exhibited multiorgan inflammation and a shortened lifespan, consistent with decreased *TGFB1* levels [49]. Nevertheless, such phenotypes were not as severe as those observed in *TGFB1* (−/−) mice. *TGFB1*-mutated mice exhibited decreased levels of active *TGFB1*, reduced TGFB signaling, and stomach tumors, suggesting that the association of LTBPs with the latent TGFB complex is important for proper *TGFB1* function in GC.

Esophageal adenocarcinoma (EAC) patients identified from the stromal molecular signature, including the CAFs marker *FAP*, had poorer outcomes, and gene ontology analysis identified a strong inflammatory component in disease progression. Key pathways included cytokine–cytokine receptor interactions and *TGFB1* signaling [35]. Gene set enrichment analysis demonstrated that the stromal signature was also relevant in the preinvasive to invasive transition of the stomach. These data implicated inflammatory pathways in the genesis of GC, which can affect prognosis.

In GC tissue samples, many of the genes with increased expression in CAFs compared with normal fibroblasts were associated with *TGFB1* activity. When CAFs were cultured in ECM, they became more motile than normal fibroblasts. GC cells incubated with CAFs were also more motile and invasive in vitro. Among the DEGs between CAFs and normal fibroblasts, *RHBDF2* (SE = 1.6) was identified [50]. *RHBDF2* knockdown (KD) in CAFs reduced their elongation and motility in response to TGFB1, whereas overexpression of *RHBDF2* in normal fibroblasts increased their motility in ECM. Intriguingly, *RHBDF2* KD in CAFs reduced cleavage of the TGFB *receptor 1* (*TGFBR1*, SE = 2.2) by *ADAM metallopeptidase domain 17* (*ADAM17* or *TACE*, SE = 1.8) and reduced expression of genes that regulate motility. Incubation of normal fibroblasts with *IL1A* (SE = 1.8), *IL1B* (SE = 5.6), or *TNFA* increased *RHBDF2* in CAFs, suggesting an important role for the TNFA-RHBDF2 axis in CAFs activation.

## 6. Conclusions

In this review article, we summarized the current understanding of the molecular features of CAFGs in GC. Intriguingly, scRNA-seq analysis clarified that the expression of all CAFGs was not necessarily restricted to CAFs alone, although many CAFGs were indeed expressed in the CAFs of GC. Thus, CAFGs may represent the activation status of CAFs. In GC, the clinicopathological features of CAFGs have been well examined, and their prognostic importance has been validated for *SPARC*, *THBS2*, *COL1A1*, *COL3A1*, *INHBA*, *PDGFC*, and *SDC2*. Nevertheless, their functional mechanisms have remained elusive compared with those in other cancers [9,10,11]. CAFGs showed synchronized expression patterns and parallel expression with *TGFB1* in CRC [9]. However, such patterns have not been validated in GC due to limitations in microdissected data. A detailed understanding of CAFGs and the TGFB pathway may be essential for developing novel therapeutic strategies for controlling GC in the near future.

## Figures and Tables

**Figure 1 cancers-17-00795-f001:**
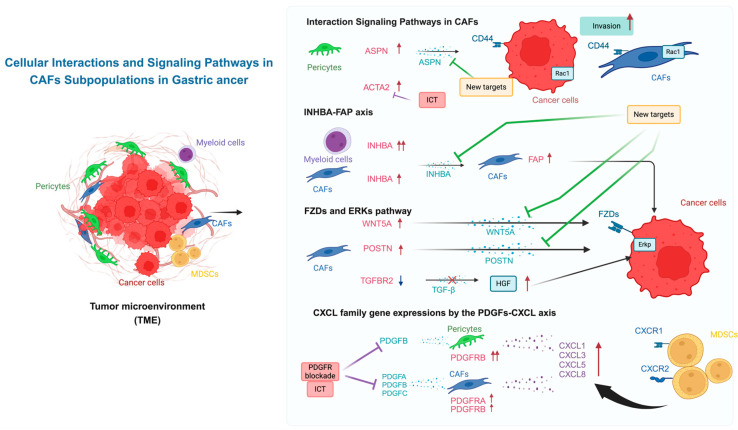
Cellular interactions and signaling pathways in CAFs subpopulations in gastric cancer.

**Figure 2 cancers-17-00795-f002:**
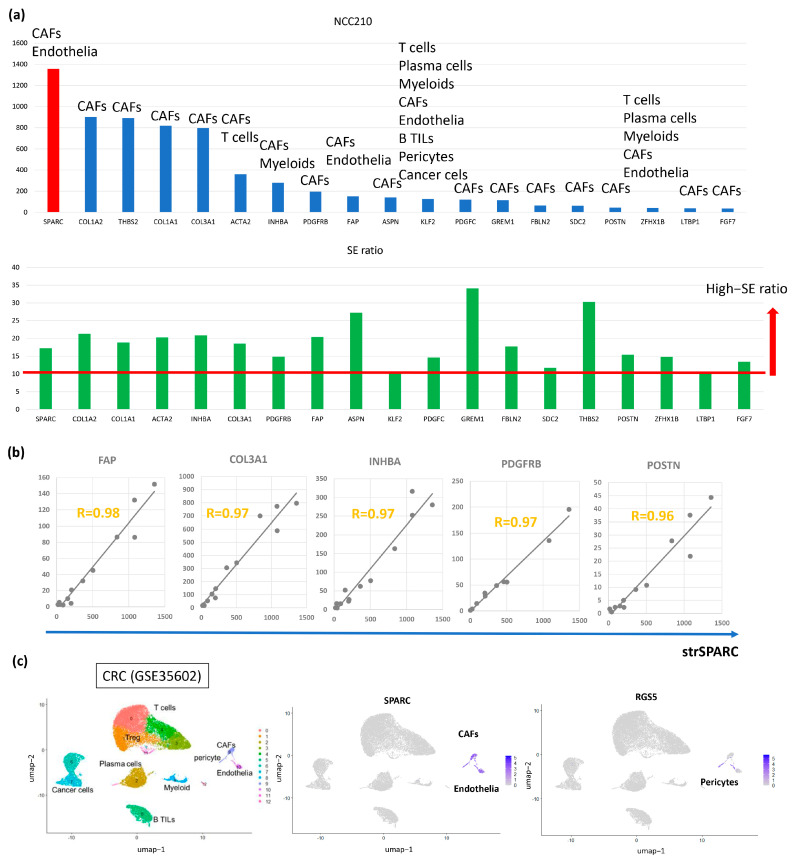
Molecular features of CAFGs identifies in CRC: (**a**) Expression amounts (upper panel) and SE ratio (lower panel) of CAFGs defined in CRC public database (GSE35602, 13 cases of CRC). scRNA-seq identified unique cell populations shown in the Upper panel. SE ratio of 10 or beyond was defined as CAFGs representing High-SE ratio (arrows). (**b**) Representative correlations of CAFGs to strSPARC are shown. Gold letters indicate R-index between SPARC and the individual CAFGs in cancer stroma of the CRC tumors (GSE35602). (**c**) scRNA-seq of the 16 CRC tumors (GSE200997) confirmed expression of *SPARC* in CAFs and endothelia. Pericyte marker, *RGS5*, was marked in subpopulations of CAFs and endothelia, respectively, suggesting that F-pericytes and E-pericytes were included in CAFs and endothelia in CRC.

**Figure 3 cancers-17-00795-f003:**
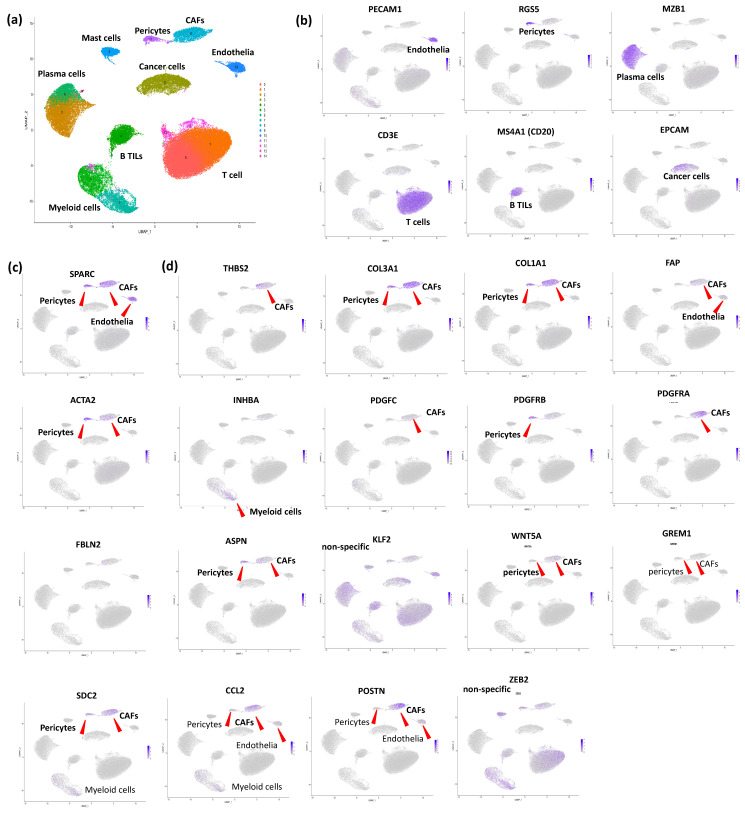
scRNA-seq identified unique cells expressing CAFGs in GC: (**a**) UMAP and (**b**) annotation of unique cell clusters by specific markers such as PECAM1 (endothelia), RGS5 (pericytes), MZB1 (plasma cells), CD3E (T cells), MS4A1 (B TILs), EPCAM (cancer cells). (**c**) scRNA-seq identified unique cells expressing SPARC as CAFs, pericytes, and endothelia. (**d**) scRNA-seq identified unique cells expressing individual CAFGs and their related genes, such as *THBS2*, *COL3A1*, *COL1A1*, *FAP*, *ACTA2*, *INHBA*, *PDGFC*, *PDGFRB*, *PDGFRA*, *FBLN2*, *ASPN*, *KLF2*, *WNT5A*, *GREM1*, *SDC2*, *CCL2*, *POSTN*, and *ZEB2*. Intriguingly, all CAFs do not express CAFGs uniquely in CAFs alone.

**Figure 4 cancers-17-00795-f004:**
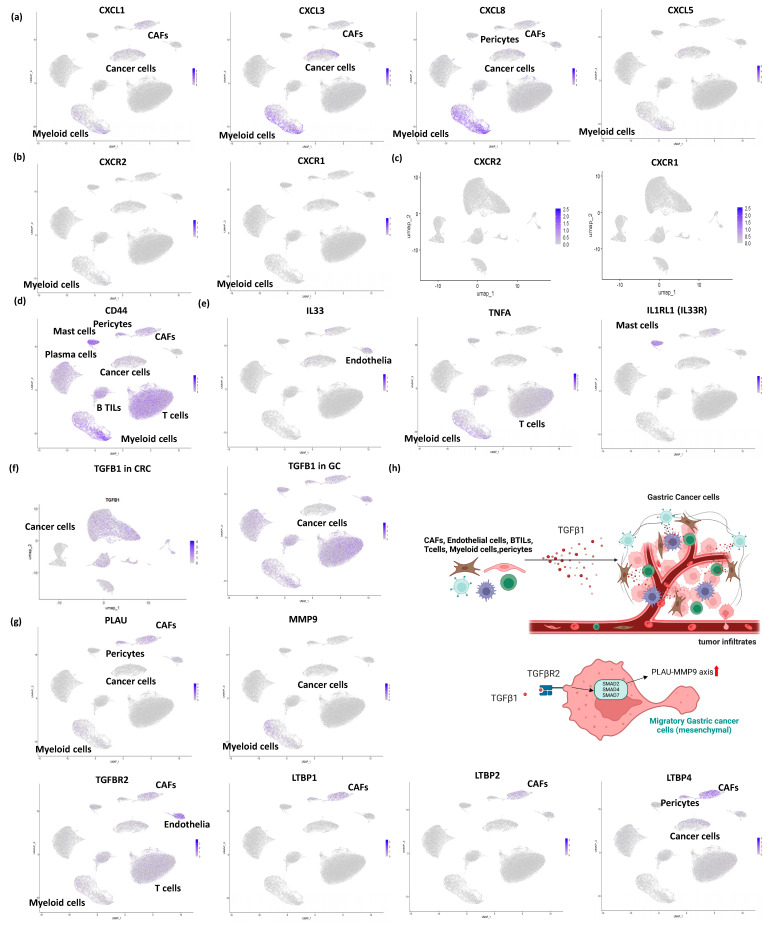
scRNA-seq identified unique cells expressing chemokines/cytokines and the related genes in GC: (**a**) CXCL family genes (*CXCL1*/*CXCL3*/*CXCL8*/*CXCL5*) are expressed rather dominantly in myeloid cells (>cancer cells > CAFs) in scRNA-seq. (**b**) Receptors (*CXCR1*/*CXCR2*) for CXCL family genes are modestly expressed in myeloid cell subpopulations in GC in comparison to CRC (**c**). (**d**) *CD44* is expressed in stromal cells rather than in cancer cells. (**e**) *IL33* is expressed dominantly in endothelia (>CAFs), while *TNFA* is expressed dominantly in myeloid cells. On the other hand, *IL33R* is expressed dominantly in mast cells. These findings proposed an important role of paracrine interactions of myeloid cells–endothelia–mast cells for GC tumor progression. (**f**) *TGFB1* is expressed dominantly in stromal cells rather than in cancer cells of GC (**left panel**) and CRC (**right panel**). (**g**) TGFβ-related genes (*PLAU*, *MMP9*, *TGFBR2*, *LTBP1*, *LTBP2*, and *LTBP4*) are confirmed in scRNA-seq in contrast to their canonical known mechanisms in the literature (**h**).

**Table 1 cancers-17-00795-t001:** Molecular features of negative prognostic CAFGs.

SYMBOL	Negative Prognostic Factor	Expression Distribution in scRNA-seq of Gastric Cancer (GSE183904)	SE Ratio in CRC
*SPARC*	Yes	CAFs, pericytes, endothelia	17.2
*THBS2*	Yes	CAFs	30.3
*COL1A1*	Yes	CAFs, pericytes, endothelia	18.9
*COL3A1*	Yes	CAFs, pericytes, endothelia	18.5
*INHBA*	Yes	CAFs < myeloid cells	20.9
*PDGFC*	Yes	CAFs (weak)	14.6
*SDC2*	Yes	CAFs, pericytes, myeloid cells	11.7

## Data Availability

In the Section 2 of the main text, we provide details about the open-source data used, the software used for analysis, and the analysis conditions. The URL for the public database is listed below. https://www.ncbi.nlm.nih.gov/geo/query/acc.cgi?acc=GSE183904, https://www.ncbi.nlm.nih.gov/geo/query/acc.cgi?acc=GSE35602, https://www.ncbi.nlm.nih.gov/geo/query/acc.cgi?acc=GSE200997, (accessed on 5 February 2025).

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
