# Peer review of "Cancer-Associated Fibroblasts Genes and Transforming Growth Factor Beta Pathway in Gastric Cancer for Novel Therapeutic Strategy"

_cancers, 2025, doi:10.3390/cancers17050795_

Round 1
Reviewer 1 Report
Comments and Suggestions for Authors
The manuscript attempt to systematically approach a topic of great interest. Authors seem to have made a great deal of effort in gathering the data, which is a positive point, but they fail, unfortunately to provide a good quality presentation, which would advance knowledge in the field.
Moreover the paper presents major flaws in methodology. It is not clear what was the strategy in the conception of the article, what software was used to process data. It seems that the paper attempt to present some data in a public available database interpreted in the light of the available literature. But then the paper is not only a review. Authors should be more clear in what they attempt to do.
The text is not at all well organized, the clinico-pathological correlations (announced in the Abstract and re-interated in the Conclusions) were not made. The entire text need to be overhauled.
The quality and the design of the figures is unacceptably poor and definitely needs to be improved. The legends do not increase the understanding of the figures.
Detailed remarks are hard to made but the authors can find some suggestions in my comments on the manuscript in the attached PDF file.

Author Response
Comments 1:
They fail, unfortunately, to provide a good quality presentation, which would advance knowledge in the field. Moreover, the paper presents major flaws in methodology. It is not clear what was the strategy in the conception of the article, what software was used to process data.
Response 1:
In order to provide a good quality presentation, we created a new “Materials and Methods” section and added references and included information of the software and databases used in the main text.
Comments 2:
It seems that the paper attempted to present some data in a public available database interpreted in the light of the available literature. But then the paper is not only a review. Authors should be clearer in what they attempt to do.
Response 2:
Our aim is to summarize the molecular characteristics of each CAFGs in gastric cancer, however CAFGs in the previous reports have bot been validated for actual expression in individual tumor components by scRNA-seq. So, we additionally included such information using the public database (GSE183904) of gastric cancer in this review paper. We included such a description on page 6, line15-18.
Comments 3:
The text is not at all well organized, the clinico-pathological correlations (announced in the Abstract and re-interated in the Conclusions) were not made. The entire text needs to be overhauled.
Response 3:
In this paper, we did not describe the clinic-pathological correlations, however we added the breakdown of GC stages is as follows: 26 primary lesions from GC patients (Stage I/II/III/IV; 3/6/14/3) in the text in the “Materials and Methods).
Comments 4:
The quality and the design of the figures is unacceptably poor and definitely needs to be improved. The legends do not increase the understanding of the figures.
Response 4:
We used Bio Render to create easy-to-understand diagrams with a high level of design as you pointed out. Moreover, we brushed up the scRNA-seq analysis in a more sophisticated manner. Concretely, we used the Seurat version 4.3.0 in R language, the number of genes per cell in the single-cell analysis ranged from 200 to 2500, and only cell populations with a mitochondrial gene ratio below 10% were included, and doublet finder was set to 7.5%, to remove doublets, and Harmony was used to reduce Batch effect. As a result, 62,519 cells were finally selected to make UMAP. We included this description on page 6, line 18-22.
Comments 5:
The legends do not increase the understanding of the figures.
Response 5:
Thank you for your suggestion. We extensively revised the legends.
Comments 6:
Detailed remarks are hard to made but the authors can find some suggestions in my comments on the manuscript in the attached PDF file.
Response 6:
Thank you for sending your PDF file. We have made corrections as you pointed out in the PDF.
Reviewer 2 Report
Comments and Suggestions for Authors
Despite the development of methods and approaches to treatment, oncological diseases still remain one of the pressing problems and challenges for modern medicine and pharmacology. A group of authors, it is especially worth emphasizing here, who are practicing doctors - surgical oncologists, collected and analyzed the literature in the field of understanding the molecular features of CAFG in GC. The study allowed us to make important general conclusions and findings on the way to a detailed understanding of CAFG and the TGFB pathway, which, if properly developed, can contribute to the development of new therapeutic strategies for GC control. I believe that this review will be of interest to a wide range of readers of the Cancers journal and fully meets the requirements of the journal.
Author Response
Comments 1:
Despite the development of methods and approaches to treatment, oncological diseases still remain one of the pressing problems and challenges for modern medicine and pharmacology. A group of authors, it is especially worth emphasizing here, who are practicing doctors - surgical oncologists, collected and analyzed the literature in the field of understanding the molecular features of CAFG in GC. The study allowed us to make important general conclusions and findings on the way to a detailed understanding of CAFG and the TGFB pathway, which, if properly developed, can contribute to the development of new therapeutic strategies for GC control. I believe that this review will be of interest to a wide range of readers of the Cancers journal and fully meets the requirements of the journal.
Response 1:
I greatly appreciate your positive comments. Thank you very much.
Reviewer 3 Report
Comments and Suggestions for Authors
This article describes the role of CAFs-associated genes and TGFB pathway in gastric cancer and discusses their role and significance as novel therapeutic targets. The authors analyse the backgroud literature and descibe their approach sufficiently. I have no notes for the authors, except a slight text editing.
Author Response
Comments 1:
This article describes the role of CAFs-associated genes and TGFB pathway in gastric cancer and discusses their role and significance as novel therapeutic targets. The authors analyze the background literature and describe their approach sufficiently. I have no notes for the authors, except a slight text editing.
Response 1:
I greatly appreciate your positive comments. Thank you very much.
Reviewer 4 Report
Comments and Suggestions for Authors
In this article, the authors reviewed the current molecular features of cancer-associated fibroblast genes (CAFGs) in gastric cancer (GC), which was crucial to the development of novel therapeutic strategies for effective GC therapy. They summarized the clinicopathological features of several CAFGs, including SPARC, THBS2, COL1A1, COL3A1, INHBA, 18 PDGFC, SDC2, and CCL2.
Here are some comments and suggestions:
1. Could they summarize the molecular features of CAFGs in GC with a table? This would give the reader a clearer picture of CAFGs in GC and easier to extract relevant information.
2. Could they introduce the potential therapeutic strategies targeted to these CAFGs in clinical or preclinical?
Author Response
Comments 1:
Could they summarize the molecular features of CAFGs in GC with a table? This would give the reader a clearer picture of CAFGs in GC and easier to extract relevant information.
Response 1:
Thank you for your pertinent comments. I made Table of the molecular characteristics of negative prognostic CAFGs in GC, including their SE ratios in CRC, prognostic significance, and expression patterns identified through scRNA-seq analysis as you pointed out.
I added the sentences on page14, line15-17
“Table 1 summarizes the molecular characteristics of negative prognostic CAFGs in GC, including their SE ratios in CRC, prognostic significance, and expression patterns identified through scRNA-seq analysis.”
Comments 2:
Could they introduce the potential therapeutic strategies targeted to these CAFGs in clinical or preclinical?
Response 2:
I added the sentences regarding therapeutic strategies targeted to these CAFGs in clinical and/or preclinical setting on page14, line17-22.
“In addition to understanding their molecular roles, we explored therapeutic strategies targeting CAFGs, in which we can potential therapeutic strategies targeting CAFGs in both clinical and preclinical settings. These strategies include sophisticated approaches such as blocking CAFG-related signaling pathways, modulating the tumor microenvironment, and targeting specific CAFG markers identified through single-cell RNA sequencing.”
Round 2
Reviewer 1 Report
Comments and Suggestions for Authors
I am satisfied with the modifications brought to the manuscript
Reviewer 4 Report
Comments and Suggestions for Authors
The authors made a great effort on the revision of this articles with improvement. All my concerns were addressed. I have no other comments to this article and recommend publishing.